# Traditional Cultures versus Next Generation Sequencing for Suspected Orthopedic Infection: Experience Gained from a Reference Centre

**DOI:** 10.3390/antibiotics12111588

**Published:** 2023-11-03

**Authors:** Sara Giordana Rimoldi, Davide Brioschi, Daniele Curreli, Federica Salari, Cristina Pagani, Alessandro Tamoni, Concetta Longobardi, Raffaella Bosari, Alberto Rizzo, Simona Landonio, Massimo Coen, Matteo Passerini, Maria Rita Gismondo, Andrea Gori, Alfonso Manzotti

**Affiliations:** 1Laboratory of Clinical Microbiology, Virology and Bioemergencies, ASST Fatebenefratelli Sacco, “L. Sacco” University Hospital, Via Giovanni Battista Grassi n. 74, 20157 Milan, Italy; rimoldi.sara@asst-fbf-sacco.it (S.G.R.);; 2Orthopedic Unit, ASST Fatebenefratelli Sacco, “L. Sacco” University Hospital, Via Giovanni Battista Grassi n. 74, 20157 Milan, Italy; 3Department of Infectious Diseases, ASST Fatebenefratelli Sacco, “L. Sacco” University Hospital, Via Giovanni Battista Grassi n. 74, 20157 Milan, Italymassimo.coen@asst-fbf-sacco.it (M.C.);; 4Department of Pathophysiology and Transplantation, University of Milan, Via Francesco Sforza n. 35, 20122 Milan, Italy; 5Centre for Multidisciplinary Research in Health Science (MACH), University of Milan, Via Francesco Sforza n. 35, 20122 Milan, Italy

**Keywords:** prosthetic joint infection, next-generation sequencing, bone infections

## Abstract

(Background) The diagnosis and the antimicrobial treatment of orthopedic infection are challenging, especially in cases with culture-negative results. New molecular methods, such as next-generation sequencing (NGS), promise to overcome some limitations of the standard culture, such as the detection of difficult-to-grow bacteria. However, data are scarce regarding the impact of molecular techniques in real-life scenarios. (Methods) We included cases of suspected orthopedic infection treated with surgery from May 2021 to September 2023. We combined traditional cultures with NGS. For NGS, we performed a metagenomic analysis of ribosomal 16s, and we queried dedicated taxonomic libraries to identify the species. To avoid false positive results, we set a cut-off of 1000 counts of the percentage of frequency of reads. (Results) We included 49 patients in our study. Our results show the presence of bacteria in 36/49 (73%) and 29/49 (59%) cases studied with NGS and traditional cultures, respectively. The concordance rate was 61%. Among the 19/49 discordant cases, in 11/19 cases, cultures were negative and NGS positive; in 4/19, cultures were positive and NGS negative; and in the remaining 4/19, different species were detected by traditional cultures and NGS. (Conclusions) Difficult-to-grow microorganisms, such as slow-growing anaerobic bacteria, were better detected by NGS compared to traditional culture in our study. However, more data to distinguish between true pathogens and contaminants are needed. NGS can be an additional tool to be used for the diagnosis of orthopedic infections and the choice of appropriate antimicrobial therapy.

## 1. Introduction

The diagnosis of orthopedic infections is often challenging. The diagnosis is based on clinical, laboratory, and radiological findings, but the microbiological results are increasingly relevant for the diagnosis to promptly set up an appropriate antibiotic therapy [1]. The traditional culture tests are considered the gold standard. For instance, some authors [2,3] proposed as diagnostic criteria two positive cultures or the presence of a fistula as a major criterion necessary to define prosthetic joint infections (PJIs). However, the culture test lacks sensitivity [4], and the cases of suspected PJI could be culture-negative in 7 to 39% of cases [5]. There are several reasons for culture failure, such as lack of wash-out antimicrobial therapy before surgery [6,7], low bacterial load, the presence of biofilms, bacteria being difficult to grow, and a transport delay from sampling to culture. Molecular biology might overcome some limits of the culture test. The available syndromic tests provide fast and easy-to-read results; however, they include a limited range of bacteria (Table 1).

Therefore, there is a growing interest in other molecular techniques, such as Next Generation Sequencing (NGS) [10]. The application of NGS platforms in microbiological laboratories in cases of culture-negative results is increasing [11,12], and the first data show excellent diagnostic accuracy, with a sensitivity and specificity of 93% and 94%, according to a recent study on synovial fluid in suspected prosthetic joint infections [13]. The advantage of this method is the possibility of identifying all the known bacterial genomic taxonomy; the sequences produced are able to identify identical bacterial communities (reads), which are aligned with databases containing all deposited bacterial genomes. However, NGS needs specialized equipment, trained microbiologists, and bioinformatics experience. With the current data, we aim to present our experience of the introduction of NGS in clinical practice for the management of suspected orthopedic infections. The primary purposes were to show the proportion of positive results from NGS in consecutive cases of suspected bone and joint infections and to present the proportion of concordant and discordant cases between NGS and traditional cultures.

## 2. Results

For this study, we included 49 patients referred to the orthopedic unit of a tertiary center in Northern Italy from May 2021 to September 2023. They were predominantly men (59%, 29/49), with a mean age of 65 (range, 15 to 87) years. The total number of samples cultured was 294. Regarding the traditional cultures, 29 out of 49 cases (59%) showed a positive result: 14 for *Staphylococcus aureus*, 8 for *Staphylococcus* spp. coagulase-negative, 2 each for Pseudomonas aeruginosa and *Corynebacterium striatum*, and 1 each for *Bacillus* spp., Enterococcus faecalis, and *Chryseobacterium* spp. In 20 out of 49 cases (41%), the traditional cultures were negative despite a suggestive clinical symptomatology. In Case 15, a patient with foot bursitis, *Staphylococcus warneri,* was only isolated in a single culture; thus, the sample was considered negative. Moreover, in Case 37, a single culture positive for *Staphylococcus cohnii* spp. cohnii was considered as a contamination (Table 2). Regarding the NGS, we obtained a positive result in 36 out of 49 (73%) cases: 10 for *Staphylococcus aureus*, 8 for *Staphylococcus* spp. coagulase-negative, 8 for *Cutibacterium acnes*, 2 each for *Pseudomonas aeruginosa*, *Bacillus* spp., and *Finegoldia magna*, and 1 each for *Streptococcus pyogenes*, *Corynebacterium striatum* (along with *Bacteroides pyogenes*), *Burkholderia* spp., and *Chryseobacterium* spp. In 13 cases, the frequency of reads (FDR) had a value < 1000, and therefore, these cases were considered negative. One case was interpreted as positive even with FDR < 1000, given that the traditional cultures were negative and the clinical suspicion of infection was high. Among the patients, 6/49 had a polymicrobial isolation: this was due to the discrepancy between the methods in four patients (Case 10, 12, 16, and 38; Table 2), while in one case (Case 35) NGS alone revealed two anaerobes and in another case (Case 36) NGS detected one bacteria not isolated with traditional cultures. The remaining 9/49 cases were negative both in NGS and in the cultures. Concordance between the two methods was achieved in 30/49 cases (61%). Among the 19 discordant cases, in 11/19 cases, cultures were negative and NGS positive; in 4/19 cases, cultures were positive and NGS negative; and in the remaining 4/19 cases, different species were detected by the techniques (Table 2).

## 3. Discussion

In our study, the proportion of cases with suspicion of orthopedic infection and with positive microbiological results were 73 and 59% for NGS and traditional cultures, respectively.

Discordant results between traditional cultures and NGS were observed in 39% of cases. Among these, in twelve cases, NGS revealed an anaerobic organism (*Cutibacterium acnes*, *Finegoldia magna*, or *Bacteroides pyogenes*) not detected in the traditional cultures. This is aligned with the available literature on the “recovery” of negative cultures, mainly due to difficult-to-culture organisms such as anaerobic bacteria [14]. There is a growing interest in the role of anaerobic bacteria in the pathogenesis of orthopedic infection, both hardware and not hardware-associated [15,16], but there is limited knowledge of the new diagnostic method for their diagnosis [17]. With our data, we believe that NGS could detect anaerobic bacteria with a better sensitivity than traditional culture media. However, more data are needed to interpret these isolations as contaminants or true pathogens. Moreover, further investigations will be necessary to better understand why bacteria such as *Burkholderia* spp. and *Streptococcus pyogenes* escaped traditional cultures and their clinical significance. One hypothesis is that slow-growing agents such as *Burkholderia* spp. may need appropriate culture media [18]; however, sometimes an a priori clinical suspicion of similar organisms is low, and the organism does not grow in traditional culture. NGS can be an effective tool for this type of scenario.

Among the thirteen cases negative for NGS, 9/13 resulted negative also in the traditional cultures. In the remaining 4/13 cases, we found discordant data with a positive culture for *Enterococcus faecalis* (*n* = 1), *Staphylococcus haemolyticus* (*n* = 1), and *Staphylococcus aureus* (*n* = 2). This is not aligned with previous results that showed a high concordance between traditional culture positive for Gram-positive bacteria and NGS (see Table 2 in [13]). Given that we collected six samples for patients, we hypothesize that these missed identifications were probably due to the use of different samples for DNA extraction and the traditional cultures. Therefore, it is of utmost importance for the next comparative studies to use the same samples for cultures and NGS.

A total of 37 out of 49 patients with orthopedic infections underwent antibiotic treatment. However, in cases with discordant results, the clinicians did not treat some of the detected bacteria considered as contaminants despite FDR > 1000. Three patients with negative traditional cultures and positive NGS for *Cutibacterium acnes* were not treated, and they obtained a clinical cure at the follow-up. It is important to note that the choice to start the treatment should take into consideration other clinical, laboratory, histological, and radiological variables and should not be based just on the microbiological results.

One of the main limitations of NGS is the absence of susceptibility tests. This can create difficulties in setting up a targeted and appropriate therapy. However, we think that NGS could help the clinician narrow the antimicrobial spectrum according to the local susceptibility patterns and not just choose to start the antimicrobial therapy. However, for some organisms, such as the aforementioned *Burkholderia* spp., the susceptibility is highly variable, and the choice of the appropriate antibiotics could be difficult without an antibiogram or the detection of the gene of resistance.

Regarding the comparison with syndromic test, it is interesting to note that NGS could identify five bacteria not present in the Unyvero Curetis panel (two *Bacillus* spp., one *Burkholderia* spp., one *Chryseobacterium* spp., and one *Bacteroides pyogenes*) and 21 bacteria not present in the BioFire Joint Infection Panel (eight *Cutibacterium acnes*, two *Bacillus* spp., six *Staphylococcus epidermidis*, one *Burkholderia* spp., one *Staphylococcus capitis*, one *Chryseobacterium* spp., one *Corynebacterium striatum*, and one *Bacteroides pyogenes*) (see Table 1 and Table 2). The absence of these microorganisms in the syndromic panels aims to minimize the false positive results since some of them, such as *Cutibacterium acnes* or *Staphylococcus epidermidis*, can also be contaminants [19]. However, there is growing interest in the pathogenic roles of these bacteria, especially in some clinical scenarios such as shoulder infections [20], hardware-associated infections [21,22], and cases of chronic and indolent infections [16]. Therefore, further studies are needed to help to distinguish between true pathogens and contaminants. Besides the necessary correlation with clinical, laboratory, histological, and radiological data, it would be promising to combine inflammatory markers in the bone with the syndromic tests and/or NGS techniques.

Despite these promising data, the duration of the method, the high costs, and the need for expertise to read and interpret the results challenge its application in routine practice in many centers. Moreover, the lack of an antibiogram and the absence of clear criteria to distinguish contaminants from true pathogens in NGS represented the major criticality within the stewardship group of our local reality. Lastly, further analyses regarding the cost-effectiveness of such an approach in different contexts are needed [23]. However, we hypothesize that the addition of NGS in the armamentarium for the diagnosis of orthopedic infections could most benefit certain categories of patients, such as those with negative traditional cultures and those on antibiotics during the sample collection. Thus, these initial data encouraged us to limit the use of NGS mainly after the results of traditional cultures became negative.

## 4. Materials and Methods

We combined the NGS 16S metagenomic analysis (Ion Torrent, Guilford, CT, USA) with traditional cultures for suspected orthopedic infections. No syndromic panel was used for this study. We collected six samples for patients for traditional culture results. For traditional cultures, all the samples collected were tested for aerobic and anaerobic bacteria on selected agar plates. The isolates were evaluated at the species level (Maldi-toff Vitek MS, Biomerieux, Marcy-l’Etoile, France) and tested for antimicrobial susceptibility with the automated analyzer Vitek 2 (Biomerieux). In the case of a possible contaminant microorganism, we considered it truly pathogenic if it was present in two or more cultures. For NGS analysis, the samples were divided into bone and cartilage materials, treated according to the manufacturer’s protocol (Invitrogen, Thermo Fisher Scientific, Waltham, MA, USA), and then pooled for DNA extraction. Metagenomic sequencing of the 16S ribosomal RNA region allowed us to investigate the seven most conserved hypervariable bacterial regions (primer set V2–4–8, 3–6, and 7–9) able to indicate the taxonomic levels of family, genus, and species. Sequences were queried against the Curated Greengenes v13.5 and MicroSEQ ID 16S Reference Library v2013.1 databases. The full 16 S kit is able to identify 107 different taxonomic genera, and the database contains all deposited bacterial genomes. Starting from a past laboratory experience [24], considering the numerous bacterial species found within a biological sample, including the non-pathogenic bone microbiota, we set a cut-off of 1000 counts of the percentage of FDR to interpret the isolated bacteria as a potential pathogen. FDR is considered the average number of reads (fragments of sequence) that align with or “cover” a known reference base to achieve bacterial identification. The IRB approval is not required for retrospective studies following the standard procedure according to the local jurisdiction.

## 5. Conclusions

In conclusion, difficult-to-grow microorganisms, such as slow-growing anaerobic bacteria, were better detected by NGS compared to traditional cultures in our study. However, more data to distinguish between true pathogens and contaminants are needed. Moreover, we state that NGS can be an additional tool to be used for the diagnosis of orthopedic infections.

## Figures and Tables

**Table 1 antibiotics-12-01588-t001:** The detected bacteria for Unyvero Curetis (OPGene, USA) and BioFire Joint Infection Panel (Biomerieux, France).

	Unyvero Curetis [8]	BioFire Joint Infection Panel [9]
**Aerobic bacteria**		
Gram-positive	*Abiotrophia defectiva* *Corynebacterium* spp. *Enterococcus faecalis* *Enterococcus* spp. *Granulicatella adiacens* *Staphylococcus aureus* *Staphylococci CONS* *Streptococcus* spp. *Streptococcus agalactiae* *Streptococcus pyogenes* *Streptococcus dysgalactiae* *Streptococcus pneumoniae*	*Enterococcus faecium* *Enterococcus faecalis* *Staphylococcus aureus* *Staphylococcus lugdunensis* *Streptococcus* spp. *Streptococcus agalactiae* *Streptococcus pyogenes* *Streptococcus pneumoniae*
Gram-negative	*Acinetobacter baumanii complex* *Citrobacter freundii/koseri* *Escherichia Coli* *Enterobacter cloacae complex* *Enterobacter aerogenes* *Klebsiella pneumoniae* *Klebsiella oxytoca* *Klebsiella variicola* *Proteus* spp. *Pseudomonas aeruginosa*	*Haemophilus influenzae* *Citrobacter* spp. *Escherichia coli* *Enterobacter cloacae complex* *Enterobacter aerogenes* *Klebsiella pneumoniae group* *Morganella morganii* *Neisseria gonorrheae* *Proteus* spp. *Pseudomonas aeruginosa* *Kingella kingae* *Salmonella* spp. *Serratia marcescens*
**Anaerobic bacteria**		
	*Bacteroides fragilis group* *Finegoldia magna* *Cutibacterium acnes*	*Bacteroides fragilis* *Anaerococcus prevotii/vaginalis* *Clostridium perfrigens* *Cutibacterium avidum/granulosum* *Finegoldia magna* *Parvimonas micra* *Peptostreptococcus anaerobius* *Peptoniphilus* spp.
**Fungi**		
	*Candida* spp. *Candida albicans* *Candida glabrata* *Candida tropicalis* *Candida krusei*	*Candida* spp. *Candida albicans*
**Genes of resistance**		
	Macrolide/Lincosamide *ermA*, *ermC* Aminoglycoside *Aac*(*6*′)*/aph*(*2*″)*aacA4* Oxacillin *mecA/mecC* Vancomycin *vanA/B* ESBL *CTX-M* Carbapenemase *IMP*, *KPC*, *NDM*, *OXA-23*, *24/40*, *48*, *58*, *VIM*	Carbapenemase *IMP*, *KPC*, *NDM*, *OXA-48-like*, *VIM* ESBL *CTX-M* Methicillin-resistance *mecA/C*, *MREJ* Vancomycin-resistance *vanA/B*

**Table 2 antibiotics-12-01588-t002:** Clinical indication for bone and/or joint biopsy and microbiological results of traditional cultures and NGS.

Case	Clinical Indications	Traditional Cultures	NGS	Concordance Methods	Discordance Methods
1	Tibial osteomyelitis	*Pseudomonas aeruginosa*	*Pseudomonas aeruginosa*	x	
2	PJI	Negative	Negative	x	
3	Septic arthritis + omeral osteomyelitis	*Staphylococcus epidermidis*	*Staphylococcus epidermidis*	x	
4	PJI	*Staphylococcus aureus*	*Staphylococcus aureus*	x	
5	Tibial osteomyelitis with hardware	Negative	*Cutibacterium acnes*		x
6	Septic arthritis	*Staphylococcus aureus*	*Staphylococcus aureus*	x	
7	PJI	*Staphylococcus aureus*	*Staphylococcus aureus*	x	
8	PJI	*Chryseobacterium* spp.	*Chryseobacterium* spp.	x	
9	PJI	Negative	Negative	x	
10	PJI	*Staphylococcus epidermidis*	*Cutibacterium* spp.		x
11	Tibial osteomyelitis	*Staphylococcus aureus*	*Staphylococcus aureus*	x	
12	Tibial osteomyelitis	*Corynebacterium coyleae*	*Bacillus* spp.	x	
13	Femoral osteomyelitis with hardware	*Staphylococcus capitis*	*Staphylococcus capitis*	x	
14	PJI	Negative	Negative	x	
15	Foot bursitis	Negative	Negative	x	
16	PJI	*Staphylococcus aureus*	*Cutibacterium acnes*		x
17	PJI	*Enterococcus faecalis*	Negative		x
18	PJI	*Bacillus* spp.	*Bacillus* spp.	x	
19	Femoral osteomyelitis	Negative	Negative	x	
20	PJI	Negative	*Streptococcus pyogenes*		x
21	Sacral osteomyelitis	Negative	Negative	x	
22	PJI	Negative	*Burkholderia* spp.		x
23	Tibial osteomyelitis with hardware	*Staphylococcus aureus*	Negative		x
24	Wrist osteomyelitis	*Staphylococcus aureus*	Negative		x
25	Foot osteomyelitis	*Staphylococcus epidermidis*	*Staphylococcus epidermidis*	x	
26	Septic arthritis	Negative	Negative	x	
27	PJI	Negative	Negative	x	
28	PJI	*Staphylococcus haemolyticus*	Negative		x
29	Femoral osteomyelitis with hardware	*Pseudomonas aeruginosa*	*Pseudomonas aeruginosa*	x	
30	Foot osteomyelitis	*Staphylococcus aureus*	*Staphylococcus aureus*	x	
31	Septic arthritis	Negative	*Cutibacterium acnes*		x
32	Foot osteomyelitis	*Staphylococcus aureus*	*Staphylococcus aureus*	x	
33	Foot osteomyelitis	*Staphylococcus aureus*	*Staphylococcus aureus*	x	
34	PJI	*Staphylococcus aureus*	*Staphylococcus aureus*	x	
35	Septic arthritis	Negative	*Finegoldia magna*, *Cutibacterium acnes*		x
36	Foot osteomyelitis	*Corynebacterium striatum*	*Corynebacterium striatum*, *Bacteroides pyogenes*		x
37	PJI	Negative	*Cutibacterium acnes*		x
38	Tibial osteomyelitis with hardware	*Staphylococcus aureus*	*Cutibacterium acnes*		x
39	PJI	Negative	Negative	x	
40	Spacer infection	*Staphylococcus aureus*	*Staphylococcus aureus*	x	
41	PJI	*Staphylococcus epidermidis*	*Staphylococcus epidermidis*	x	
42	Septic arthritis	Negative	*Staphylococcus hominis*		x
43	PJI	*Staphylococcus epidermidis*	*Staphylococcus epidermidis*	x	
44	PJI	Negative	*Staphylococcus epidermidis*		x
45	Septic osteonecrosis	Negative	*Cutibacterium acnes*		x
46	PJI	Negative	*Finegoldia magna*		x
47	PJI	Negative	*Cutibacterium acnes*		x
48	Tibial osteomyelitis	*Staphylococcus aureus*	*Staphylococcus aureus*	x	
49	PJI	*Staphylococcus epidermidis*	*Staphylococcus epidermidis*	x	
Total				30/49 (61%)	19/49 (39%)

NGS, next-generation sequencing; PJI, prosthetic joint infection.

## Data Availability

Data are available upon request.

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
