# Peer review of "Traditional Cultures versus Next Generation Sequencing for Suspected Orthopedic Infection: Experience Gained from a Reference Centre"

_antibiotics, 2023, doi:10.3390/antibiotics12111588_

Round 1

Reviewer 1 Report

Comments and Suggestions for Authors

This well written paper presents interesting data on the diagnosis of orthopaedic infections confronting cultural and molecular tests of materials collected from cases of suspected prosthetic Joint infections. Minor points could be improved:

1.       In Tab 1 the molecular tests of Unyvero Curetis and Biofire Joint Infection Panel are compared; it would be better to add even the respective data about the genetic antibiotic resistance markers.  Moreover, a comment about the different bacteria panels chosen by Curetis and bioMerieux would be welcomed.

2.       The frequency of reads (FDR), as a semi-quantitative result of the molecular tests, could be better presented since most clinicians are used to simple qualitative result of positive or negative parameter.

3.        The discordance between cultural and molecular methods seems to favor identification of Staphylococcus aureus and Enterococcus faecalis in traditional cultures while Cutibacterium, Finegoldia, Streptococcus and Bulkholderia in molecular tests. How can be explained the superiority of traditional cultures for Staphylococcus and Enterococcus?

4.       In Material and Methods is not stated if Unyvero Curetis or Biofire Joint Infection bioMerieux or both were used in the present Study.

Author Response

Dear Editor and Reviewers,

We thank you for your helpful and constructive comments on the submitted version of our manuscript and for giving us the opportunity to improve it.

All comments and suggestions have been taken into account as indicated in the provided detailed point-by-point response. All changes made in the revised version of our manuscript are tracked as requested.

REVIEWER 1

This well written paper presents interesting data on the diagnosis of orthopaedic infections confronting cultural and molecular tests of materials collected from cases of suspected prosthetic Joint infections. Minor points could be improved:

  1. In Tab 1 the molecular tests of Unyvero Curetis and Biofire Joint Infection Panel are compared; it would be better to add even the respective data about the genetic antibiotic resistance markers.  Moreover, a comment about the different bacteria panels chosen by Curetis and bioMerieux would be welcomed.

      Response: we agree it is important to also report the genes of resistance, which were added in the revised table. We also added a sentence in the discussion regarding the choice fo Unyvero Curetis and Biofire to exclude some microorganisms from their panel. Please, find the sentence below (in the text you can also find the references):

       The absence of these microorganisms in the syndromic panels aims to minimize the false positive results, since some of them, such as Cutibacterium acnes or Staphylococcus epidermidis, can also be contaminant. However, there is growing interest in the pathogenic roles of these bacteria, especially in some clinical scenarios such as shoulder infections, hardware-associated infection, and cases of chronic and indolent infections”.

       Regarding NGS, the primary aim of the implementation of this technique in clinical practice was to identify microorganism, not the mechanism of resistance. To note, all our isolates were tested with standard antibiogram and none of them reported resistance.

  1. The frequency of reads (FDR), as a semi-quantitative result of the molecular tests, could be better presented since most clinicians are used to simple qualitative result of positive or negative parameter.

      Response: we agree with the reviewer that FDR deserves a more extensive explanation and that most of the clinicians are used to interpret a microbiological result as positive or negative. In the Materials and Methods section, we report the following sequence to increase the clarity of the methodology:

      “FDR is considered as the average number of reads (fragments of sequence) that align to, or "cover," known reference base to achieve bacterial identification”.

  1. The discordance between cultural and molecular methods seems to favor identification of Staphylococcus aureus and Enterococcus faecalis in traditional cultures while Cutibacterium, Finegoldia, Streptococcus and Bulkholderia in molecular tests. How can be explained the superiority of traditional cultures for Staphylococcus and Enterococcus?

       Response: this point was an intriguing matter of discussion also among us. At the beginning we could not find a reasonable explanation, since no previous study showed this discordance, and the biological plausibility was missing. However, we extensively talked with the Micro lab and the Ortho team, and we hypothesized that this was a problem of samples’ collection. Given that all the patients included in the study had six intra-operative samples, it could be possible that in some cases the samples used for traditional cultures were different from the samples used for NGS. The lesson we learned is to use the same samples for the two techniques to avoid inconsistency. We report it in the discussion section as follows:

      “Among the twelve cases negative for NGS, 8/12 resulted negative also in the traditional cultures. In the remaining 4/12 cases, we found discordant data with a positive culture for: Enterococcus faecalis (n=1), Staphylococcus haemolyticus (n=1) and Staphylococcus aureus (n=2). This is not aligned with previous results which showed a high concordance between traditional culture positive for Gram-positive bacteria and NGS (see Table 2 in [18]). Given that we collected six samples for patients, we hypothesize that these missed identifications were probably due to the use of different samples for the DNA extraction and the tra-ditional cultures. Therefore, it is of utmost importance for next comparative studies to use the same samples for cultures and NGS”.

  1. In Material and Methods is not stated if Unyvero Curetis or Biofire Joint Infection bioMerieux or both were used in the present Study.

       Response: thank you for this comment of methodology. We added a sentence in the Material and Methods section to clarify this point.

Reviewer 2 Report

Comments and Suggestions for Authors

-Why were Cutibacterium acnes not diagnosed on Culture but not NGS.

- Could you provide sensitivity and specificity values from other studies?

-Are the results only valid for orthopedic studies?

-Given the high cost can you provide some studies that speak about Cost Benefit Analysis of NGS.

Author Response

Dear Editor and Reviewers,

We thank you for your helpful and constructive comments on the submitted version of our manuscript and for giving us the opportunity to improve it.

All comments and suggestions have been taken into account as indicated in the provided detailed point-by-point response. All changes made in the revised version of our manuscript are tracked as requested.

REVIEWER 2

  1. Why were Cutibacterium acnes not diagnosed on Culture but not NGS.

Response: Cutibacterium acnes is an anaerobic microorganism. Therefore, it needs appropriate carriage and preservation of the sample to increase the chance to isolate it because the contact with the air (and, therefore, oxygen, could limit its growth in traditional cultures). Please, find the relative comment to this interesting topic in the discussion section:

“Discordant results between traditional cultures and NGS were observed in 31% of cases. Among these, in five cases NGS revealed an anaerobic organism (Cutibacterium acnes or Finegoldia magna) not detected in the traditional cultures. This is aligned with the available literature on the “recovery” of negative cultures, mainly due to difficult-to-cultures organism such as anaerobic bacteria”.

  1. Could you provide sensitivity and specificity values from other studies?

Response: we added a sentence in the Introduction section, which you can find below:

“…and the first data show an excellent diagnostic accuracy, with a sensitivity and specificity of 93% and 94% according to a recent study on synovial fluid in suspected prosthetic joint infections”

  1. Given the high cost can you provide some studies that speak about Cost Benefit Analysis of NGS.

Response: this is an extremely interest comment. There is paucity of data for this topic. In the Discussion section report the need to have more data and we also report the results of one study as a reference.

Lastly, further analyses regarding the cost-effectiveness of such an approach in different context are needed”. Reference: Torchia MT, Austin DC, Kunkel ST, Dwyer KW, Moschetti WE. Next-Generation Sequencing vs Culture-Based Methods for Diagnosing Periprosthetic Joint Infection After Total Knee Arthroplasty: A Cost-Effectiveness Analysis. J Arthroplasty. 2019 Jul;34(7):1333-1341. doi: 10.1016/j.arth.2019.03.029. Epub 2019 Mar 19. PMID: 31005439.

Reviewer 3 Report

Comments and Suggestions for Authors

Overall the topic is good and informative. Introduction needs moe explanation and  there are still ore facts which should be described like purpose and why study was conducted. The sample size is small give justification. results are insufficient so I sugest add more results. 

Author Response

Dear Editor and Reviewers,

We thank you for your helpful and constructive comments on the submitted version of our manuscript and for giving us the opportunity to improve it.

All comments and suggestions have been taken into account as indicated in the provided detailed point-by-point response. All changes made in the revised version of our manuscript are tracked as requested.

REVIEWER 3

Overall the topic is good and informative. Introduction needs moe explanation and  there are still ore facts which should be described like purpose and why study was conducted. The sample size is small give justification. results are insufficient so I sugest add more results. 

Response: we thank the reviewer since this comment allowed us to expand the results by including patients studied with NGS until September 2023. This permitted to increase the number of consecutive patients from 35 to 49. The results remained quite similar, but now they are more consistent. We also followed the suggestion to report the primary purposes in the Introduction section.

Round 2

Reviewer 2 Report

Comments and Suggestions for Authors

- Thanks for your quick response to my suggestions.

- Thanks for providing the sensitivity and specificity of the test. It is high which is good.

- Given the high cost in which scenarios do you suggest using it over more conventional methods?